# Exploring FeLV-Gag-Based VLPs as a New Vaccine Platform—Analysis of Production and Immunogenicity

**DOI:** 10.3390/ijms24109025

**Published:** 2023-05-19

**Authors:** Raquel Ortiz, Ana Barajas, Anna Pons-Grífols, Benjamin Trinité, Ferran Tarrés-Freixas, Carla Rovirosa, Victor Urrea, Antonio Barreiro, Anna Gonzalez-Tendero, Maria Cardona, Laura Ferrer, Bonaventura Clotet, Jorge Carrillo, Carmen Aguilar-Gurrieri, Julià Blanco

**Affiliations:** 1AIDS Research Institute, IrsiCaixa, Campus Can Ruti, 08916 Badalona, Spainjcarrillo@irsicaixa.es (J.C.); 2Doctorate School, Microbiology Department, Universitat Autònoma de Barcelona, 08193 Bellaterra, Spain; 3Doctorate School, Medicine Department, University of Vic-Central University of Catalonia (UVic-UCC), 08500 Vic, Spain; 4HIPRA, 17170 Amer, Spain; 5Infectious Diseases Department, Germans Trias I Pujol Hospital, 08916 Badalona, Spain; 6CIBERINFEC, ISCIII, 28029 Madrid, Spain; 7Germans Trias I Pujol Research Institute (IGTP), Campus Can Ruti, 08916 Badalona, Spain

**Keywords:** FeLV vaccination, virus-like particle, humoral immunity, glycogag, veterinary science

## Abstract

Feline leukemia virus (FeLV) is one of the most prevalent infectious diseases in domestic cats. Although different commercial vaccines are available, none of them provides full protection. Thus, efforts to design a more efficient vaccine are needed. Our group has successfully engineered HIV-1 Gag-based VLPs that induce a potent and functional immune response against the HIV-1 transmembrane protein gp41. Here, we propose to use this concept to generate FeLV-Gag-based VLPs as a novel vaccine strategy against this retrovirus. By analogy to our HIV-1 platform, a fragment of the FeLV transmembrane p15E protein was exposed on FeLV-Gag-based VLPs. After optimization of Gag sequences, the immunogenicity of the selected candidates was evaluated in C57BL/6 and BALB/c mice, showing strong cellular and humoral responses to Gag but failing to generate anti-p15E antibodies. Altogether, this study not only tests the versatility of the enveloped VLP-based vaccine platform but also sheds light on FeLV vaccine research.

## 1. Introduction

Feline leukemia virus (FeLV), an enveloped RNA virus belonging to the genus *Gammaretrovirus* of the *Retroviridae* family [1,2], was first identified in the early 1960s by Jarret et al. [3,4]. Last decade’s prevalence of FeLV infection in cats ranged between 1% and 5.5% depending on the location (USA, Canada, or Southern Europe) [5,6,7,8,9]. Even though the prevalence has decreased in the past 20 years due to testing, isolation, and vaccination programs, FeLV is still one of the most common and important pathogens in domestic cats worldwide [10,11]. FeLV infection can present with different clinical signs: neoplasia, such as lymphomas and leukemia; and non-neoplastic diseases, such as anemia, enteritis, and secondary infections attributable to immunosuppression [12,13,14,15,16]. Humoral and cellular immune responses have been associated with protection from FeLV infection and disease; however, the mechanisms that determine the infection outcome have yet to be fully resolved [1].

FeLV is considered a simple retrovirus with three genes: *gag*, *pol*, and *env*. The *env* gene encodes the Env viral envelope glycoprotein, which is responsible for the specific attachment of the virus to host target cells. It consists of two proteolytically cleaved subunits: gp70 (surface, SU, 70 kDa) and p15E (transmembrane, TM, 15 kDa). Both subunits are targeted by the humoral response to generate protective neutralizing antibodies [17,18,19,20]. However, following infection and proviral integration into the host’s cell genome, the magnitude of viral replication and clinical progression seems to be defined primarily by the cellular response [21,22].

Since FeLV’s discovery, there have been many efforts to design a vaccine able to induce both humoral and cellular protective responses [21,22,23]. Currently, three vaccine formulations against FeLV are commercially available: (i) classical inactivated virus vaccines [24,25]; (ii) subunit vaccines based on FeLV recombinant antigens [26]; and (iii) infectious recombinant canarypox virus engineered to express FeLV genes [27,28]. However, they all fail to provide full protection against FeLV infection, most likely due to their inability to induce a potent neutralizing immune response [29,30,31,32,33]. Therefore, new efforts are needed to design more efficient vaccines that protect against infection.

Virus-Like Particles (VLPs) represent an emerging vaccine platform that may enhance both humoral and cellular immune responses, making them a good vaccine candidate against FeLV. VLPs mimic the structure of native viruses but are non-replicative since they lack the viral genome. VLPs can carry antigens in particulate form and with repetitive surface patterns, making them efficient immunogens [34,35,36,37]. Particularly, VLPs based on the human immunodeficiency virus (HIV) are nanoparticles wrapped by a lipid bilayer and can be generated by the sole expression of the HIV structural Gag protein [38,39]. Our group developed a novel approach by directly fusing an antigen of interest to the N-terminal region of HIV-Gag. This strategy allowed the generation of VLPs with a high density of immunogen on their surface, which could induce a potent and functional immune response even in the absence of adjuvants and at a low VLP dose [40,41]. Therefore, retroviral VLPs could represent an excellent alternative to more conventional vaccine platforms.

Here, we propose to generate FeLV-based VLPs as a novel vaccine strategy against this retrovirus. By analogy to our HIV strategy and considering that p15E is one of the targets of neutralizing antibodies [19,20,42,43], a fragment of this protein was exposed on the surface of HIV-based VLPs. As an alternative, we also designed FeLV-Gag-based VLPs loaded with the same immunogen, which may elicit not only humoral responses against p15E but also against FeLV-Gag [21]. After FeLV-Gag-based VLP optimization, the immunogenicity of the selected candidates was evaluated in two different animal models, C57BL/6 and BALB/c mice, showing a humoral and cellular response against FeLV-Gag but no response against the TM antigen.

## 2. Results

### 2.1. Generation of HIV and FeLV-Based Virus-like Particles

Our group has developed an HIV-based VLP vaccine platform in which a small fragment of the HIV envelope glycoprotein gp41 subunit is fused via its transmembrane domain to the N-terminal part of the HIV-Gag protein [41]. This strategy allows for a high-density display of the antigen on the surface of VLPs (a 1:1 ratio with the Gag protein; about 2500 copies of HIV-Gag are estimated per VLP particle [44]). To study the versatility of our platform and adapt it as a FeLV-based VLP vaccine, we selected p15E of the FeLV envelope glycoprotein as the antigen to be presented on the surface of the VLPs.

To characterize FeLV-based VLPs and compare them to HIV-based VLPs, two different recombinant proteins based on each retrovirus were designed. A fragment of the p15E protein (residues 542 to 610) of the FeLV envelope glycoprotein, including the membrane-spanning domain (MSD), was fused to either HIV-Gag (TM-HIV) or FeLV-Gag (TM-FeLV) (Figure 1A). Both fusion proteins contained a signal peptide (SP) and a Flag Tag at the N-terminus (Figure 1A). Additionally, two control fusion proteins containing the SP, the Flag Tag, the MSD, and the respective Gag but lacking the extracellular part of the p15E fragment were designed as controls for immunization experiments (Figure 1A).

Expression of recombinant fusion proteins in transiently transfected Expi293F cells was analyzed by western blot using antibodies targeting the TM antigen (Flag) and Gag protein (HIV p24 or FeLV p27). The results showed a band of the expected molecular weight (asterisks in Figure 1B, P lanes), indicating that the full protein was properly expressed (TM-HIV = 67.1 kDa; TM-FeLV = 69.2 kDa). Remarkably, the recombinant proteins were hardly detected in the supernatant, except for FeLV-Gag containing proteins (Figure 1B, SN lanes), indicating that the fusion proteins might be retained inside the cells and not secreted to the extracellular media. Besides the whole fusion protein band, several low-molecular-weight specimens were detected, probably due to protein processing or degradation. Low molecular weight fragments were differentially identified in both constructs, suggesting that Gag from HIV and FeLV were not equally processed (Figure 1B). Thus, a 35 kDa fragment was detected in the TM-FeLV cell lysate using the anti-Flag antibody. However, this protein was not observed in the TM-HIV lane (Figure 1B). In line with that, when TM-FeLV was analyzed with an anti-p27 antibody, a clear band of 42 kDa was present in the supernatant of the cell culture (Figure 1B), probably corresponding to the other part of the cleaved protein observed with the anti-Flag antibody. This observation was confirmed when the two control fusion proteins lacking the p15E protein were also transiently transfected into Expi293F cells. Both proteins were expressed at their expected molecular weights (Control-HIV = 62.1 kDa; Control-FeLV = 64.3 kDa, Figure 1C). Similar to Figure 1B, when developed with an anti-p27 antibody, a prominent 42 kDa band was observed in the supernatant of the Control-FeLV fusion protein, reinforcing the idea of the proteolytic cleavage of Gag (Figure 1C).

To evaluate the localization of the expressed proteins, transiently transfected Expi293F cells were stained at the cellular surface with an anti-Flag antibody and intracellularly with anti-HIV p24 or anti-FeLV p27 antibodies. The results showed that the extracellular fragment of the fusion protein, including the TM immunogen, was barely exposed on the surface of Gag-positive cells (Figure 1D). Instead, a much stronger signal was detected when the p27 staining was performed intracellularly, further supporting the intracellular retention of all fusion proteins (Figure 1D). Accordingly, a higher amount of HIV p24 and FeLV p27, determined by ELISA, was detected in cell lysates than supernatants of transiently transfected cells (83% in TM-HIV and 74% in TM-FeLV) (Figure 1E).

To confirm the intracellular retention of all fusion proteins, we evaluated by TEM whether VLP assembly and budding were occurring intracellularly. Both FeLV- and HIV-based VLPs were found to accumulate in the perinuclear area at the rough endoplasmic reticulum (RER) of producing cells, showing a similar morphology and diameter (Figure 2). The confirmation of the presence of intracellular VLPs led us to adapt a VLP extraction protocol from Titchener-Hooker et al. [45,46]. The extracted material was visualized by cryo-EM, showing again spherical VLPs with a similar morphology among all of them and a slightly higher diameter for HIV-based VLPs (120 nm) than for FeLV-based VLPs (100 nm) (Figure 2).

Taken together, these data confirmed that both TM-HIV and TM-FeLV VLPs are produced but retained inside the cells. In addition, a large proportion of FeLV-Gag proteins undergo truncation, which could be related to the presence of proteolytic cleavage sites associated with glycogag formation during the FeLV life cycle [47].

### 2.2. Sequential Deletion of Gag to Reduce Proteolytic Cleavage

To reduce TM-FeLV processing, we tested two different signal peptides: human GMCSF (MWLQSLLLLGTVACSIA) and an immunoglobulin-derived SP (MDWTWRFLFVVAAATGVQS) (Appendix A, TM-FeLV_SP2). No differences in VLP production, cellular localization, or VLP morphology were noticed (Appendix A), suggesting that the signal peptide had no impact on protein processing.

FeLV, as well as other gammaretroviruses, such as MuLV and KoRV [48,49], generates an alternative form of the Gag polyprotein by an alternative translation of the initiation codon [48], resulting in a secreted and glycosylated form of the Gag protein (glycogag) [50]. Upon translation, FeLV glycogag is translocated to the plasma membrane of the infected cell, where it is cleaved and a C-terminal product of 40 kDa is released [51]. To reduce the possible proteolytic cleavage of FeLV-Gag and considering that matrix (MA or p15) is dispensable for retroviral particle formation [52,53,54], we designed three novel fusion proteins showing N-terminus Gag deletions (TM-ΔMA, TM-Δ153, and TM-ΔMA-Δp12). In addition, since HIV p6 is located in the C-terminus of Gag and has a similar function to FeLV p12 [55,56], we designed an additional construct in which the p12 subunit was shifted to the C-terminus (TM-ΔMA-p12CT) (Figure 3A).

Western blot analysis of cell lysates and culture supernatants of transiently transfected Expi293F cells showed that all four proteins were expressed at their expected molecular weights (TM-ΔMA = 54.6 kDa; TM-Δ153 = 51.9 kDa; TM-ΔMA-p12CT = 54.6 kDa; TM-ΔMA-Δp12 = 47.3 kDa) (Figure 3B). Faint 20 and 15 kDa bands were identified using anti-Flag antibodies in TM-∆MA and TM-∆153 cell lysate fractions, respectively (Figure 3B). However, a 42 kDa band was detected with the anti-p27 antibody in the supernatant of the cell cultures (Figure 3B). Hence, these results suggest that the deletion of the first 153 amino acids of FeLV-Gag is not sufficient to avoid protein cleavage and glycogag formation. In contrast, we neither detected any cleavage band in TM-ΔMA-p12CT or TM-ΔMA-Δp12 cell lysates using an anti-Flag antibody nor the 42 kDa band on the cell culture supernatant when developed with an anti-p27 antibody (Figure 3B).

We also evaluated the possible retention of the fusion proteins inside the cells by flow cytometry. (Figure 3C). Flag was mainly detected when intracellular staining was applied, suggesting that all the fusion proteins were retained inside the cells and were not reaching the plasma membrane, similarly to TM-FeLV (Figure 3C). Quantification of p27 by ELISA showed that the different fusion proteins were mainly accumulated inside the cells (>70% in all cases) (Figure 3D), reinforcing our previous observation that the fusion proteins are retained in the cells.

Further characterization by Cryo-EM of extracted VLPs from transiently transfected Expi293F cells showed that TM-ΔMA and TM-Δ153 generated VLPs with similar morphology to TM-FeLV (Figure 2D and Figure 3D). However, no VLPs were detected for the TM-ΔMA-p12CT fusion protein, and heterogenous VLPs were observed for TM-ΔMA-Δp12 (Figure 3D). Taken together, these data indicate that the deletion of the Gag cleavage site impacts VLP formation.

### 2.3. Improving FeLV-Based VLP Production

Previous results suggest that the FeLV-Gag cleavage site is located in MA or p12 sequences. To identify the protease responsible for the cleavage of FeLV-Gag as well as its putative cleavage sites, we analyzed the FeLV-Gag protein sequence using the MEROPS database [57]. Several cleavage sites were identified, and ADAMDEC1 protease was noted as the putative protease responsible for the cleavage at positions 198–201 of our recombinant protein [58,59]. Furthermore, we identified a late domain in this region corresponding to residues 126–130 in sequence LYPVL that could be the target of the viral protease [60,61]. Therefore, new fusion proteins were designed: (1) TM-VA, with two mutations L199V and R200A in FeLV-Gag; (2) TM-ΔProR, deletion of the late domain region (FeLV-Gag residues 128 to 153); (3) TM-ΔProR-VA, deletion of the late domain site plus the double mutation; (4) TM-ΔMA-VA, deletion of the full matrix plus the double mutation; and (5) TM-Δ153-VA, deletion of the region containing the first 153 aa of FeLV-Gag plus the double mutation (Figure 4A).

Expression of recombinant fusion proteins in transiently transfected Expi293F cells was analyzed by western blot, showing specific bands at the expected molecular weight (TM-VA = 69.2 kDa; TM-ΔProR = 66.5 kDa; TM-ΔProR-VA = 66.5 kDa; TM-ΔMA-VA = 54.6 kDa; TM-Δ153-VA = 51.9 kDa). All recombinant proteins were identified in cell lysates and supernatants (Figure 4B, red asterisks). Even though all modified FeLV-Gag fusion proteins showed a 42 kDa band in the supernatant when revealed with anti-p27 (FeLV), only TM-VA and TM-ΔProR-VA did not show a clear band of around 30 kDa present in TM-FeLV and TM-ΔProR or around 20 kDa present in TM-ΔMA-VA and TM-Δ153-VA when the cell lysate was revealed with anti-Flag antibody (Figure 4B, blue asterisks). This suggests that TM-VA and TM-ΔProR-VA reduce the cleavage of FeLV-Gag.

All fusion proteins examined were retained inside the cells, as suggested by Figure 4C, in which intracellular staining with anti-p27 and anti-Flag antibodies on transiently transfected Expi293F cells showed similar levels of expression (around 25% of double positive cells detected). In addition, all cell cultures presented more than 60% of p27 in the cell lysate when quantified by in-house ELISA (Figure 4D), while a slightly lower percentage was observed for TM-ΔProR (53%). To confirm the formation and presence of FeLV-based VLPs in cell extracts, cryo-EM was performed on all extracted samples (Figure 4D) and used as a decision criterion to evaluate VLP presence and morphology. Even though FeLV-based VLPs were observed for all fusion proteins, TM-ΔProR-VA and TM-VA showed a more homogeneous morphology.

Taken together, these data suggest that TM-∆ProR-VA and TM-VA fusion proteins promote the expression of uncleaved, morphologically correct intracellular FeLV-based VLPs. Therefore, we retained both as vaccine candidates for subsequent analyses.

### 2.4. Immunogenicity of TM-FeLV VLPs in C57BL/6 Mice

To evaluate the immunogenicity of FeLV-based VLPs, C57BL/6 mice were electroporated intramuscularly with plasmids coding for the fusion proteins TM-HIV, TM-FeLV, TM-VA, and TM-∆ProR-VA, and the controls: Control-HIV and Control-FeLV. Animals were vaccinated with two doses of plasmid DNA (20 µg/dose) at weeks 0 and 3, and the end point of the experiment was set at week 6 post-first immunization (Figure 5A).

First, we evaluated the humoral response against the TM antigen using an in-house ELISA. Neither HIV- nor FeLV-based VLP vaccinated animals generated detectable antibodies against TM (Figure 5B). To understand these results, we tested the possible immunodominant role of the Flag tag in the fusion protein; however, no humoral response against this epitope was detected in any group. Control-HIV and TM-HIV were included in the analysis as negative controls, and as expected, no humoral response was generated in these groups (Figure 5C). Next, we analyzed the humoral response against Gag proteins. Analysis of the response against FeLV p27 showed a homogeneous signal in all FeLV-based VLP-vaccinated groups. All groups reached a plateau after a single dose of electroporated DNA, which was maintained three weeks after the second immunization (Figure 5D). No statistically significant differences were observed among the FeLV-based VLP-vaccinated groups. Similarly, the humoral response against HIV p24 was analyzed by an in-house ELISA assay, and it followed similar kinetics, reaching comparable levels in the TM-HIV and Control-HIV groups (Figure 5E). Very low or no responses against FeLV-Gag were observed in HIV-based VLP-vaccinated animals and vice versa, confirming the specificity of antibody quantification (Figure 5D,E).

Finally, the elicitation of cellular responses against TM, p27 (FeLV), and p24 (HIV) was evaluated by an IFNɣ ELISpot assay using cryopreserved splenocytes from all vaccinated groups. Consistent with the lack of humoral responses against TM, none of the VLPs, neither FeLV nor HIV, were able to generate a detectable T-cell response against this protein, but both were able to generate a T-cell response against FeLV-Gag (p27 protein) or HIV-Gag (p24 protein), respectively (Figure 5F).

Altogether, these data indicate that both HIV-based and FeLV-based VLPs generated humoral and cellular responses against their respective Gag proteins. In contrast, neither humoral nor cellular responses were observed against the TM antigen.

### 2.5. Immunogenicity of TM-FeLV VLPs in BALB/c Mice

To evaluate whether the lack of generation of immune responses against TM in C57BL/6 was strain-dependent, we also tested BALB/c mice, a model that tends to produce a stronger humoral response than the C57Bl/6 strain [62]. BALB/c mice were immunized in a VLP/VLP regime in combination with adjuvants to enhance immune responses (Figure 6A).

FeLV-based VLPs were extracted from transiently transfected Expi293F cells and purified by double cushion sucrose ultracentrifugation (Appendix A). Purified material was then analyzed by NTA, which allowed for quantification (ranging from 1.27 × 10^10^ to 1.7 × 10^10^ particles/mL) and calculation of particle size (ranging from 155 nm to 180 nm in diameter) (Appendix A). Western blot evaluation of purified fractions confirmed the presence of both TM and Gag fragments in all fusion proteins at their expected molecular weights (Appendix A).

In a first set of experiments, the TM-ΔProR-VA vaccine candidate was formulated with different adjuvants: (i) MPLA + Al(OH)_3_, (ii) QuilA + cholesterol +DDABr, (iii) squalene-based adjuvant (SQBA), and (iv) QuilA + Al(OH)_3_. A synthetic TM peptide (residues 541 to 582 of Env FeLV) formulated with MPLA + Al(OH)_3_ and a non-adjuvanted TM-ΔProR-VA VLP preparation were also included as controls. BALB/c mice were immunized subcutaneously at the hock following a homologous VLP regimen of two doses, at weeks 0 and 3. The experimental endpoint was set at week 6 post-first immunization (Figure 6A).

We evaluated the humoral response against the TM antigen, showing that no humoral immune response against the TM antigen was induced in any of the tested conditions (Figure 6B). Even though all vaccine formulations increased the antibody titer against p27 after the second immunization, the TM-ΔProR-VA vaccine candidate formulated with SQBA and MPLA + Al(OH)_3_ induced a statistically significant difference compared to the non-adjuvanted group (Figure 6C). Further, T cell responses against TM and p27 were evaluated by IFNɣ ELISpot using cryopreserved splenocytes from all vaccinated groups (Figure 6D). None of the tested adjuvants was able to enhance T cell responses against TM. However, a T-cell immune response was generated against p27 in MPLA + Al(OH)_3_ and QuilA + cholesterol +DDABr groups. Taken together and regardless of the immune response against TM antigen, the highest humoral and cellular immune responses against p27 were generated by the animals vaccinated with TM-ΔProR-VA formulated with MPLA + Al(OH)_3_. Consequently, we have kept MPLA + Al(OH)_3_ as the adjuvant combination to test in further experiments.

As FeLV-based VLPs are produced in Expi293F cells, a human cell line, we expected that Expi293F proteins (of human origin) would be incorporated into the VLPs and could be immunogenic in mice. The analysis of the humoral response against Expi293F cell proteins was performed for all experimental groups by flow cytometry (Appendix A). The data suggest that the adjuvants inducing a higher humoral response against p27 are also the ones that induce higher titers of antibodies against the Expi293F cells, specifically the SQBA. Animals immunized with the synthetic TM peptide served as negative controls.

Once the optimal adjuvant was selected, we screened the different VLP candidates. In a second experiment, BALB/c mice were immunized subcutaneously following the previous protocol (Figure 6A), with three FeLV-based VLPs: TM-FeLV, TM-VA, and TM- ΔProR-VA (Appendix A) adjuvanted with MPLA + Al(OH)_3_; a non-adjuvanted TM-ΔProR-VA was included as a control. Again, the humoral response against TM was negative for all fusion proteins (Figure 6E). However, antibody titers against p27 were detectable in all vaccinated groups (Figure 6F). Statistically significant differences were observed between TM-ΔProR-VA adjuvanted with MPLA + Al(OH)_3_ and the other experimental groups (Figure 6F), confirming that this is the most immunogenic preparation. Analysis of the humoral response against Expi293F cell proteins was performed by flow cytometry (Appendix A), showing a similar response between the three adjuvanted VLP preparations. In contrast, non-adjuvanted VLPs elicited a weak response against the Expi293F cells.

To sum up, homologous TM-∆ProR-VA VLPs immunization regimens in BALB/c mice successfully induced humoral responses against p27, especially when MPLA + Al(OH)_3_ or SQBA are used as adjuvants; however, this formulation fails to induce immune responses against the FeLV TM immunogen, confirming the poor immunogenicity of this Env fragment.

## 3. Discussion

Several vaccines are available for preventing FeLV infection, but none provide full protection [13]. Therefore, new vaccine platforms are needed for the development of a successful FeLV vaccine. Enveloped Gag-based VLPs mimic the virus morphology and, therefore, may display viral immunogens in a more natural manner and at a higher density, enhancing their immunogenicity [63,64]. Several VLP-based vaccines for humans are commercially available, including vaccines against Human Papilloma Virus (HPV) [65] and Hepatitis B Virus (HBV) [66]. Several other VLP-based vaccines are currently undergoing preclinical and clinical development [67]. Here we propose an innovative design for a FeLV-based VLP vaccine. This is, to our knowledge, the first study of a FeLV-Gag-based VLP vaccine.

Recently, we have developed a high-density platform of HIV-based VLPs where the antigen of interest is exposed on the surface of the VLPs by fusing an HIV-1 immunogen to Gag through a transmembrane domain and a linker. Considering these results [41], we hypothesized that our VLP platform could be versatile enough to be adapted as a vaccine for other retroviruses. Particularly, we focused on the FeLV retrovirus, not only because there is room for improvement regarding the commercial vaccines available but also because previous descriptions of neutralizing FeLV antibodies directed against the TM subunit of the Env glycoprotein were similar and comparable to those described for HIV [68]. In that way, FeLV is not only of interest to veterinary practice but could also be a relevant model to help understand HIV immunology.

First, we decided to use p15E of FeLV Env as the immunogen to be presented on the VLP surface since this protein is targeted by neutralizing antibodies. [19,20,42,43]. A fragment of the p15E protein was fused to HIV-Gag or to FeLV-Gag. After characterization of these fusion proteins by western blot, flow cytometry, and ELISA, we concluded that the designed fusion proteins were successfully produced, even though VLPs were retained inside the cells and did not reach the extracellular compartment. Moreover, we noticed that a large proportion of FeLV-Gag-based fusion proteins underwent truncation, as a band of 42 kDa was observed in the supernatant of transfected cells. We hypothesized that this instability may originate from the existence of a glycosylated form of Gag in FeLV, a remarkable difference between FeLV and HIV-Gag proteins. It has been described that FeLV has two starting codons in Gag. One of them encodes for the polyprotein precursor of the internal capsid protein, and the other encodes for an alternative type II transmembrane glycoprotein form, known as glycogag, which has an expected molecular weight of 80 kDa and a soluble cleavage product of 40 kDa [47,69,70,71,72]. Glycogag is dispensable for virus replication but appears to be an important virulence determinant [48]. It is involved in a late step of viral budding or assembly [73,74]. Our fusion proteins are structurally similar but probably dissimilar to glycogag. Therefore, the potential expression of our immunogen as type II transmembrane proteins (suggested by the presence of large amounts of a 42 kDa Gag fragment in cell culture supernatant) could have a detrimental impact on FeLV-Gag-based VLP production. To avoid this potential effect, we explored different strategies to stabilize FeLV-Gag, first by sequential deletions of several Gag domains and then by analyzing in silico the proteolytic cleavage site. We performed several N-terminus modifications, such as deletion of specific regions of MA or p12, and introduced mutations to avoid the proteolytic cleavage site. TEM confirmed that our fusion proteins were capable of producing VLPs, but they were retained in the cell. Nevertheless, FeLV-Gag-based VLPs could be successfully extracted from transiently transfected Expi293F cells. Microscopy images were used as a selection criterion to identify the best vaccine candidates and test them in in vivo models.

Immunogenicity studies were performed in C57BL/6 and BALB/c mice with two different regimens: two doses of DNA or two doses of purified FeLV-based VLPs. Moreover, different adjuvants were added to enhance immune responses induced by FeLV-based VLPs. A comparison between the VLP regimen and the DNA regimen showed similar profiles of humoral responses. Although high antibody titers against FeLV-Gag were elicited, no antibodies against TM were detected. The absence of humoral and cellular responses against TM could be due to the small size of the protein, improper epitope expression on the surface of the VLPs, or the immunosuppressive function of p15E, although the reported immunosuppressive domain was excluded from our immunogen [75,76,77,78]. Moreover, it remains possible that C57bl/6 and Balb/c mouse models may have TCR or MHC repertoires unable to recognize the TM peptide, which is known to elicit immunogenic responses in rabbit [20,42] and rat [79] models.

In vivo experiments demonstrate that FeLV-Gag-based VLPs are immunogenic when administered as DNA vaccines or as purified VLPs. From all fusion proteins designed and tested, we conclude that the best fusion protein producing VLPs and inducing a good humoral and cellular response is TM-∆ProR-VA. Even if high titers of antibodies are elicited without adjuvant when animals are immunized with DNA, we tested different adjuvants to see if they could induce a higher cellular response. Two adjuvant formulations, MPLA + Al(OH)_3_ and SQBA, elicited higher titers of antibodies against p27 FeLV-Gag. However, SQBA is a squalene-based oil-in-water, and we do not know how this could affect VLP’s morphology; moreover, based on cellular immune response, we can observe that the best formulation includes MPLA + Al(OH)_3_ in the final vaccine. Taken together, our results confirm the versatility of our VLP platform and its ability to adapt to other retroviruses, such as FeLV. Even though no response against the surface antigen loaded on the FeLV-based VLPs was generated, biologically relevant humoral and cellular responses against p27 (the FeLV-Gag capsid) were elicited. For this reason, it is necessary to explore other FeLV antigens while maintaining the FeLV-Gag-based VLPs as a vaccine platform.

As there is no response against the antigen on the surface, future perspectives could include trying to generate new FeLV-Gag-based VLPs presenting more complex immunogens, such as larger antigens, which are characterized by their ability to induce better immune responses against other neutralizing epitopes. As well, testing the TM FeLV-based VLPs in other animal models, such as rabbits or rats, could also be informative. However, cellular responses against Gag are reported [21,22], reinforcing that our FeLV-VLPs might be good vaccine candidates as we elicited cellular responses against FeLV-Gag.

## 4. Materials and Methods

### 4.1. Plasmids

All DNA sequences were synthesized at GeneArt (ThermoFisher Scientific, Waltham, MA, USA) and cloned into the pcDNA3.4-TOPO vector (ThermoFisher Scientific, Waltham, MA, USA). In addition, they were subcloned into the pVAX1 vector (ThermoFisher Scientific, Waltham, MA, USA) using FastDigest *KpnI* and *XhoI* restriction enzymes (ThermoFisher Scientific, Waltham, MA, USA). All plasmids were transformed into One Shot TOP10 Chemically Competent *E. coli* (Invitrogen) for plasmid DNA amplification. Plasmids were purified in endotoxin-free conditions using the ZymoPure II Plasmid Maxiprep Kit (Zymo Research) and sterile filtered at 0.22 µm (Millipore). Nucleic acid concentration was measured using NanoDrop One/One (ThermoFisher Scientific, Waltham, MA, USA), based on the absorbance at 260 nm.

### 4.2. Cell Line, Culture Conditions, and Transfection

The Expi293F cell line (ThermoFisher Scientific, Waltham, MA, USA) was used for protein and VLP production. Cells were cultured in Expi293 Expression Medium (Gibco) at 37 °C, 8% CO_2_, and under agitation at 125 rpm. All transfections were performed using the ExpiFectamine transfection kit (Gibco) following the manufacturer’s recommendation. Cells and supernatants were harvested 48 h after transfection.

### 4.3. Cell Disruption, VLP Extraction, and Purification

A method to recover intracellular VLPs was adapted from Titchener-Hooker, N. et al. [45,46]. The cell pellet was resuspended in 1 pellet volume (PV) of lysis buffer at 4 °C (20 mM), Phosphate buffer pH 7.4 (Merck, Rahway, NJ, USA), 2 mM EDTA (ThermoFisher Scientific, Waltham, MA, USA), 2 mM EGTA (Merck, Rahway, NJ, USA), and Protease Inhibitor (Complete™ ULTRA Tablets EDTA-free, Merck, Rahway, NJ, USA).

Cell disruption was carried out by manual homogenization with a tissue grinder (CS1, KIMBLE) for 1 min on ice. After cell disruption, 2 PV of lysis buffer supplemented with 0.2% Triton X-100 were mixed for 4 h at 4 °C to ensure VLP release. Centrifugation at 3000× *g* for 15 min was carried out to remove cellular debris and contaminants. The supernatant was incubated with Amberlite XAD-4 beads (Merck) for 2 h at 4 °C for the removal of Triton X-100. Triton-free supernatant was recovered after centrifugation at 800× *g* for 5 min.

Supernatants containing extracted VLPs were filtered through a 0.22 µm pore size (Millipore) for sterility. Samples were further purified by ultracentrifugation in a 70% and 30% double sucrose cushion at 40,000× *g* for 2.5 h. Sucrose was removed from the sample by dialysis with Spectra-Por Float-A-Lyzer G2 (Merck) following the manufacturer’s recommendation against 1× PBS. The final sucrose concentration in VLP vaccine preparation was expected to be lower than 5%.

### 4.4. Analysis of VLP and Protein Production

#### 4.4.1. Western Blotting

Samples (15 µg of total protein) were boiled for 5 min at 95 °C and subjected to electrophoresis in NuPAGE Bis-Tris 4% to 12% (Thermo Fisher Scientific, Waltham, MA, USA). Proteins were transferred onto a PVDF membrane (Bio-Rad) using the Trans-Blot Turbo Transfer System (Bio-Rad). Membranes were blocked for 1 h at room temperature with blocking buffer (5% (*w*/*v*) non-fat skim milk powder in 1× PBS and 0.05% Tween20). Membranes were incubated overnight at 4 °C with the primary antibodies: anti-FeLV p27 monoclonal antibody [clone PF12J-10A] (1:2000, Abcam), anti-Flag monoclonal antibody [clone FG4R] (1:1000, ThermoFisher Scientific, Waltham, MA, USA), or anti-HIV p24 monoclonal antibody [clone 39/5.4A] (1:2000, Abcam). After washing, incubation with the secondary antibody, HRP-conjugated AffiniPure Donkey anti-mouse IgG (H + L) (1:10,000, Jackson ImmunoResearch, West Grove, PA, USA), was done for 1 h. Membranes were developed using SuperSignal West Pico PLUS Chemiluminescent Substrate (ThermoFisher Scientific, Waltham, MA, USA) or SuperSignal West Femto Maximum Sensitivity Substrate (ThermoFisher Scientific, Waltham, MA, USA), depending on the band’s signal and according to the manufacturer’s protocol.

#### 4.4.2. Flow Cytometry Analysis of VLP Expression

Transiently transfected Expi293F cells were analyzed by flow cytometry. VLP-producing cells were stained with APC-labeled anti-DYKDDDDK Tag Antibody [clone L5] (1:500, BioLegend, San Diego, CA, USA), FITC-labeled anti-HIV p24 Gag monoclonal antibody [clone KC57] (1:200, Beckman Coulter), or FITC-labeled anti-FeLV p27 polyclonal antibody (1:100, ThermoFisher Scientific, Waltham, MA, USA). For cell surface protein expression, cells were stained with anti-Tag antibody (15 min at room temperature), then fixed and permeabilized with FIX&PERM (Invitrogen) (15 min at room temperature), and stained with the specific anti-Gag antibody (30 min at room temperature). For intracellular staining, cells were fixed and permeabilized (15 min at room temperature) before incubation with antibodies. Cells were acquired using a BD FACS LSRII Flow Cytometer with BD FACSDiva Software version 8.0.1.1. The flow cytometry results were analyzed using FlowJo™ v10.6.1 Software (BD).

#### 4.4.3. Enzyme-Linked Immunosorbent Assay (ELISA) for VLP Quantification

FeLV-based VLP were quantified using an in-house-developed anti-p27 sandwich ELISA. Nunc Maxisorp ELISA plates (ThermoFisher Scientific, Waltham, MA, USA) were coated with 100 µL of anti-FeLV p27 goat polyclonal antibody (1µg/mL) (Abcam, Cambridge, UK) overnight at 4 °C. Then, plates were blocked using 1× PBS, 1% bovine serum albumin (BSA, Miltenyi Biotech, Cologne, Germany), and 0.05% Tween20 (Merck, Rahway, NJ, USA) for 2 h at room temperature. Recombinant feline leukemia virus p27 (ProSpec) was used as standard. Samples were treated with 0.1× RIPA Buffer (Cell Signaling Technology, Danvers, MA, USA) for 1 h and added (100 µL) to the plate. Overnight incubation was performed at 4 °C. Following that, incubation with an anti-FeLV p27 monoclonal antibody (PF12J-10A) (1:1000, Abcam, Cambridge, UK) was used as the primary antibody for 2 h at room temperature. HRP-conjugated AffiniPure goat polyclonal anti-mouse IgG (1:10,000, Jackson ImmunoResearch, West Grove, PA, USA) was used as a detection antibody. Plates were revealed with o-Phenylenediamine dihydrochloride (OPD) (Sigma Aldrich, St. Louis, MO, USA) and stopped using 2N of H_2_SO_4_. The signal was analyzed as the optical density (OD) at 492 nm with noise correction at 620 nm. For the quantification of HIV p24 in HIV-based VLPs, the INNOTEST HIV Antigen mAb (Fujirebio, Tokyo, Japan) assay was performed according to the manufacturer’s protocol.

Quantification of Gag content was performed in cellular pellets and culture supernatants. The fraction of Gag in both preparations was calculated.

#### 4.4.4. Transmission (TEM) and Cryo-Transmission Electron Microscopy (Cryo-EM)

Cells producing VLPs were analyzed by transmission electron microscopy (TEM). Briefly, transiently transfected Expi293F cells were fixed with 2.5% glutaraldehyde in PBS 0.1 M for 2 h at 4 °C, post-fixed with 1% osmium tetroxide with 0.8% potassium ferrocyanide for 2 h, and dehydrated in increasing concentrations of ethanol. Then, pellets were embedded in epon resin and polymerized at 60 °C for 48 h. Sections of 70 nm in thickness were obtained with a Leica EM UC6 microtome (Wetzlar), stained with 2% uranyl acetate and Reynold’s solution (0.2% sodium citrate and 0.2% lead nitrate), and analyzed using a JEM-1400 transmission electron microscope (Jeol Ltd., Akishima, Japan). All images were taken at 120 kV.

Purified VLP preparations were analyzed by Cryo-EM. VLPs were deposited on a carbon-coated copper grid and prepared using a Leica EM GP workstation (Leica). VLPs were observed with a Jeol JEM-2011 (Jeol Ltd., Akishima, Japan), equipped with a CCD 895 USC4000 camera (Gatan, Pleasanton, CA, USA).

#### 4.4.5. Nanoparticle Tracking Analysis (NTA)

The quantification of vesicles in each vaccine vial was performed by NTA to assess both particle size and concentration [80] using a Nanosight NS300 instrument (Malvern Panalytical, Malvern, UK). Vesicles were resuspended in 1× PBS and diluted to the working range of the system (10^6^–10^9^ particles/mL). Videos were captured using a sCMOS camera and analyzed with the Nanosight NS300 software (version 3.4).

### 4.5. Mice Immunization and Immunogenicity Analyses

All experimental procedures were done under Spanish and European laws and the Institutional Animal Care and Ethics Committee of the Center for Comparative Medicine and Bioimage (CMCiB, Badalona, Spain). They were performed by trained researchers and approved by the regional authorities (Generalitat de Catalunya, Authorization ID: 10583 and ID: 11617). All experimental protocols were performed following the principles of the 3Rs, prioritizing the welfare of the animals used in the research.

DNA immunization was performed in groups of ten six-week-old C57Bl/6JOlaHsd mice (Envigo). Males and females were equally represented in each group. Two doses, at weeks 0 and 3, of sterile endotoxin-free DNA were electroporated intramuscularly at the hind leg (20 µg DNA in physiological saline) using a NEPA21 Electroporation System (Nepagene). The electroporation protocol consisted of 8 pulses of 20 ms with a 1-s interval at 60 V. Prior to each immunization, blood samples were collected via facial vein puncture. Serum was recovered from whole blood after coagulation, centrifugation for 10 min at 4000× *g,* and heat inactivation for 30 min at 56 °C. Euthanasia of all animals was performed at week 6, when blood samples and spleens from each animal were taken for ex vivo immune analysis.

Immunizations with purified VLPs (ranging from 1.27 × 10^8^ to 1.7 × 10^8^ particles/dose) or TM peptide (20 µg/dose) were performed in groups of ten six-week-old BALB/cOlaHsd mice (Envigo, Indianapolis, IN, USA). Males and females were equally represented in each group. Two doses, at weeks 0 and 3, of purified VLPs were injected at the hock. Different adjuvants were tested; the final concentration in the vaccine formulation was 0.5 mg/mL MPLA (Merck, Rahway, NJ, USA) +1 g/L Al(OH)_3_ (Vertellus, Indianapolis, IN, USA); 20 µg/mL QuilA (Desert King, San Diego, USA) +1.55 g/L cholesterol (Gibco) +2 g/L DDABr (TCI); 500 g/L SQBA (Hipra, Catalunya, Spain); and 10 µg/mL QuilA (Desert King, San Diego, USA) +3 g/L Al(OH)_3_ (Vertellus, Indianapolis, IN, USA). Adjuvants were mixed at a ratio of 1:1 with VLP preparations. Sample collection and processing were performed as described for the DNA electroporation experiment.

### 4.6. Evaluation of Humoral Response by ELISA

The levels of antibodies against HIV-Gag, FeLV p27, and FeLV p15E in mouse serum samples were determined by an in-house sandwich ELISA.

For anti-HIV-Gag and anti-FeLV p27 antibody quantification, Nunc MaxiSorp 96-well plates (ThermoFisher Scientific, Waltham, MA, USA) were coated with 100 ng/well of recombinant Gag p55 [41] or FeLV p27 (ProSpec) and incubated overnight at 4 °C. Then, coated plates were blocked with PBS, 1% bovine serum albumin (BSA, Miltenyi Biotech, Cologne, Germany), and 0.05% Tween20 (Merck, Rahway, NJ, USA) for 2 h at room temperature. Mouse anti-HIV p24 monoclonal antibody (clone 39/5.4A, Abcam, Cambridge, UK) or mouse anti-p27 monoclonal antibody (clone PF12J-10A, Abcam, Cambridge, UK) were used as standards for HIV-Gag and FeLV p27 determination, respectively. Blocking buffer-diluted serum samples (1:100 and 1:1000) were added and incubated overnight at 4 °C. Total bound IgG was determined with a secondary HRP-conjugated AffiniPure goat polyclonal anti-mouse IgG (1:10,000, Jackson ImmunoResearch, West Grove, PA, USA). Plates were developed with o-Phenylenediamine dihydrochloride (OPD, Sigma Aldrich, St. Louis, MO, USA) and stopped using 2N of H_2_SO_4_ (Sigma Aldrich, St. Louis, MO, USA). The signal was analyzed as the optical density (OD) at 492 nm with noise correction at 620 nm.

For anti-p15E antibody quantification, a recombinant human Fc-fusion protein containing the p15 peptide in the N-terminal region (TM-huIgG) was used as an antigen. Nunc MaxiSorp 96-well plates (ThermoFisher Scientific) were coated with 100 ng/well of pure goat anti-human IgG Fc (Jackson ImmunoResearch, West Grove, PA, USA) and incubated overnight at 4 °C. Then, plates were blocked using PBS, 1% BSA, and 0.05% Tween20 for 2 h at room temperature. After that, plates were incubated with TM-huIgG for 2 h at room temperature. Mouse anti-human IgG_1_ Fc (Merck, Rahway, NJ, USA) was used as standard. The rest of the method followed the same scheme as for anti-Gag HIV-Gag and anti-FeLV p27.

#### Design and Production of TM-huIgG

The TM-huIgG used for p15E quantification is a fusion protein designed in-house. We fused the p15E fragment (residues 541 to 582 of Env FeLV) to the huIgG Fc portion at the hinge region [81]. It was cloned using FastDigest *KpnI* and *NheI* restriction enzymes. The supernatant of transiently transfected Expi293F cells was used directly for the in-house ELISA. The fusion protein was characterized by Western blot using an HRP-AffiniPure goat polyclonal anti-human IgG (Jackson ImmunoResearch, West Grove, PA, USA) and a rabbit polyclonal anti-TM antibody and quantified by ELISA (30 µg/mL) as described elsewhere [81] (Appendix A). The proper exposure of the TM fragment was further confirmed by ELISA using rabbit polyclonal anti-TM antibodies and HRP-AffiniPure goat polyclonal anti-rabbit IgG (Jackson ImmunoResearch, West Grove, PA, USA) (Appendix A).

### 4.7. Evaluation of Cellular Immune Responses by ELISpot

Splenocytes from vaccinated animals were seeded at 0.4 × 10^6^ cells/well in ELISpot white PVDF plates (Merck, Rahway, NJ, USA) precoated with 0.2 µg/well of anti-mouse IFN-γ antibodies (AN18; Biolegend, San Diego, CA, USA) and blocked with 10% FBS-supplemented RPMI (R10) (Gibco). Cells were stimulated overnight at 37 °C and 5% CO_2_ with either recombinant p27 protein at 14 µg/mL (ProSpec), p15E peptide at 14 µg/mL (residues 541 to 582 of FeLV-A Env), or a pool of 10 overlapping Gag 15-mer peptides (covering residues 314–412 of HIV-1 Env) at a concentration of 14 µg/mL per peptide. Concanavalin A (ConA; 7 µg/mL; Merck, Rahway, NJ, USA) and R10 alone were used as positive and negative controls, respectively. IFN-γ secretion was detected with an anti-mIFN-γ biotinylated-mAb (clone R4-6A2, 1:2000, Biolegend, San Diego, CA, USA) and streptavidin-AP (1:2000, Mabtech, Nacka Strand, Sweden) and developed with the AP Conjugate Substrate Kit (Bio-Rad) following the manufacturer’s protocol.

### 4.8. Evaluation of Anti-Expi293 Humoral Responses by Flow Cytometry

The presence of antibodies targeting proteins at the surface of Expi293F cells in immunized mouse sera was determined by flow cytometry with a BD FACS LSRII Flow Cytometer. In short, Expi293F cells were incubated with mouse sera samples (1:1000) and stained with AlexaFluor647 goat anti-mouse IgG Fc (1:400, Jackson Immunoresearch, West Grove, PA, USA). Flow cytometry results were analyzed using FlowJo™ v10.6.1 Software (BD).

### 4.9. Statistical Analysis

Statistical analyses were performed using Prism 9.0 (GraphPad Software Inc., Poznań, Poland) and R v4.1.1. Comparisons in immunogenicity were tested, including undetectable data (under LOD), using the Peto-Peto rank test for cross-sectional comparisons and the Petro-Pentrice generalized Wilcoxon test for longitudinal ones. Multiple comparisons were adjusted using the Benjamini and Hochberg method (FDR). For all analyses, a *p*-value of less than or equal to 0.05 was considered significant.

## Figures and Tables

**Figure 1 ijms-24-09025-f001:**
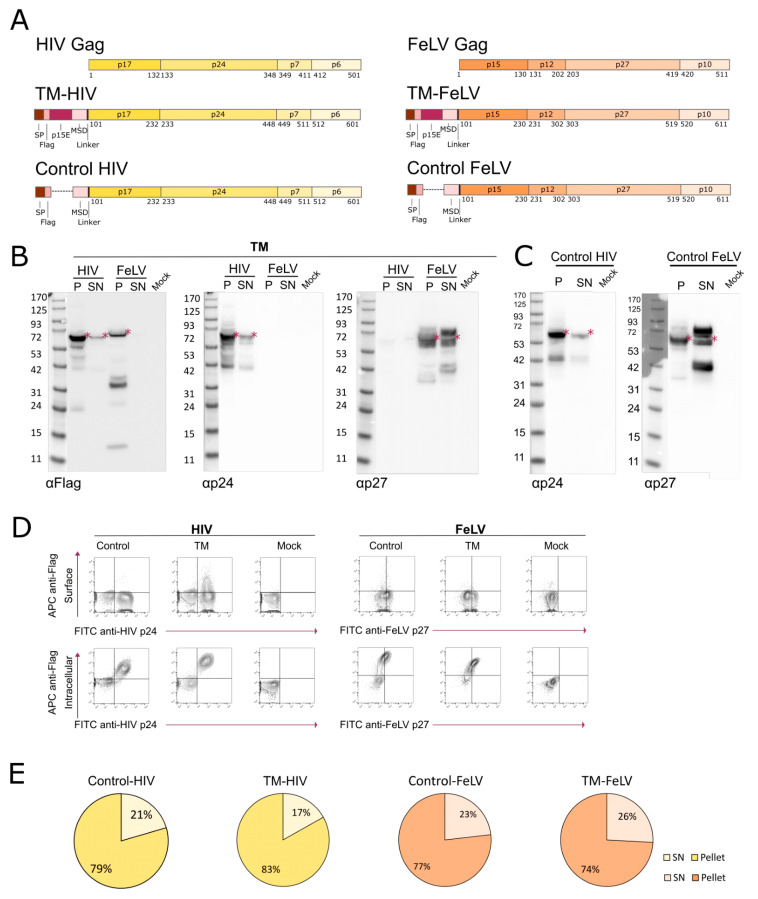
**Design and characterization of HIV and FeLV fusion proteins.** (**A**) Schematic representation of the constructs coding for TM-based VLPs with HIV-Gag or FeLV-Gag. A control fusion protein for each Gag is also represented. SP: Signal Peptide; MSD: Membrane-Spanning Domain. (**B**) Western blot analysis of TM-based fusion proteins transiently transfected in Expi293F cells. Red asterisks indicate the expected molecular weight of fusion proteins. P: Cell lysate; SN: Supernatant. (**C**) Western blot analysis of Control fusion proteins transiently transfected in Expi293F cells. Asterisks indicate the expected molecular weight of fusion proteins. (**D**) Representative flow cytometry panels for fusion protein expression. Upper panel: cell surface detection of Flag-tag and intracellular staining of Gag in transfected cells. Lower panel: identification of intracellular Flag Tag and Gag. (**E**) Pie charts of p24 and p27 levels in cell lysates and supernatants of the transiently transfected Expi293F cells.

**Figure 2 ijms-24-09025-f002:**
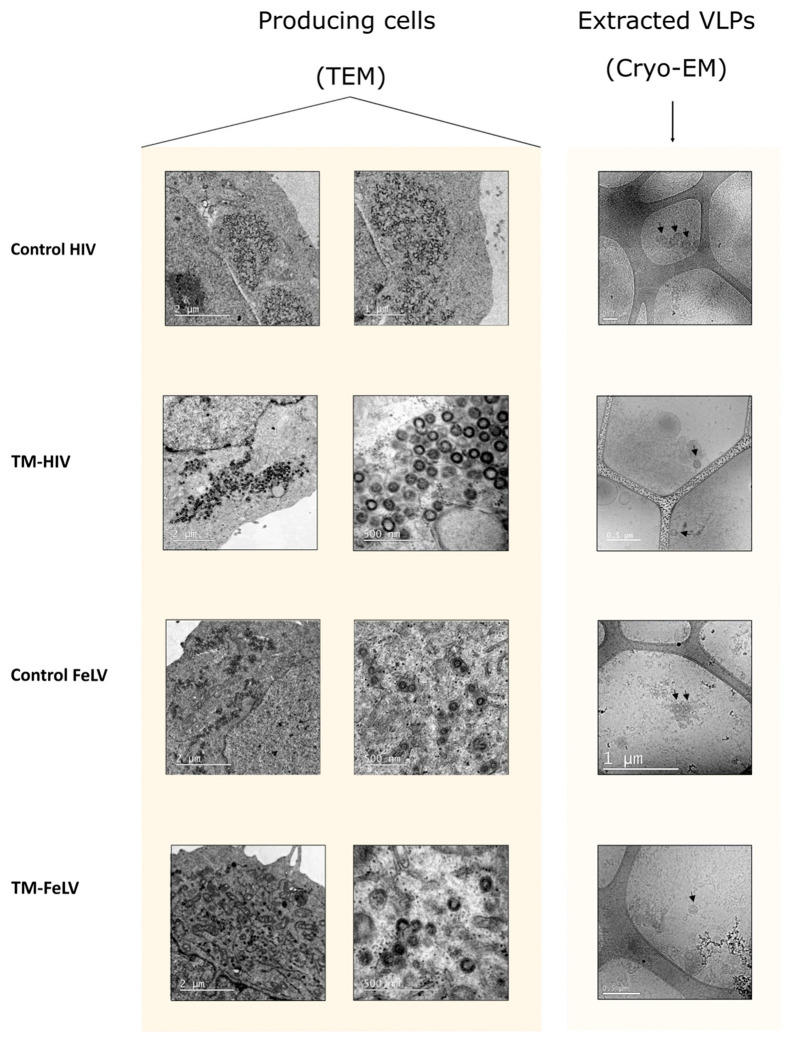
**TEM and Cryo-EM of control and TM-Gag fusion proteins.** TEM images were obtained from transiently transfected Expi293F cells expressing the indicated VLPs. Cryo-EM images were obtained using purified VLPs.

**Figure 3 ijms-24-09025-f003:**
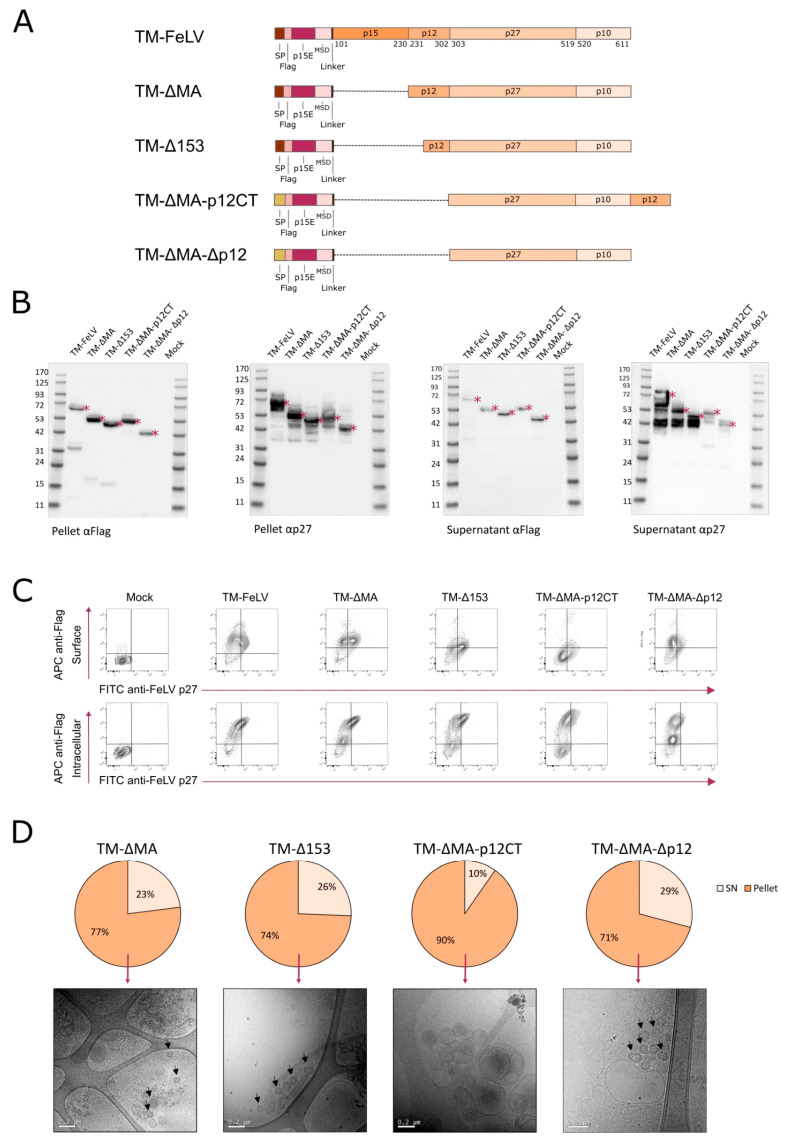
**Design and characterization of TM-Gag fusion proteins lacking gag subdomains.** (**A**) Schematic representation of the modifications for each fusion protein. The screening was performed using two sets of signal peptides (as indicated). SP: Signal Peptide; MSD: Membrane-Spanning Domain. (**B**) Western blot was developed with anti-Flag or anti-p27 antibodies to analyze the expression of TM-based fusion proteins in transiently transfected Expi293F cells. Red asterisks indicate the expected molecular weight of fusion proteins. P: Cell Lysate; SN: Supernatant. (**C**) Representative flow cytometry panels for fusion protein expression. Upper panel: identification of Flag Tag on the surface of the cells and intracellular Gag. Lower panel: identification of intracellular Flag Tag and Gag. (**D**) p27 quantification of cell lysates and supernatants of transiently transfected Expi293F cells. Cryo-EM images of purified VLPs from cell pellets are shown.

**Figure 4 ijms-24-09025-f004:**
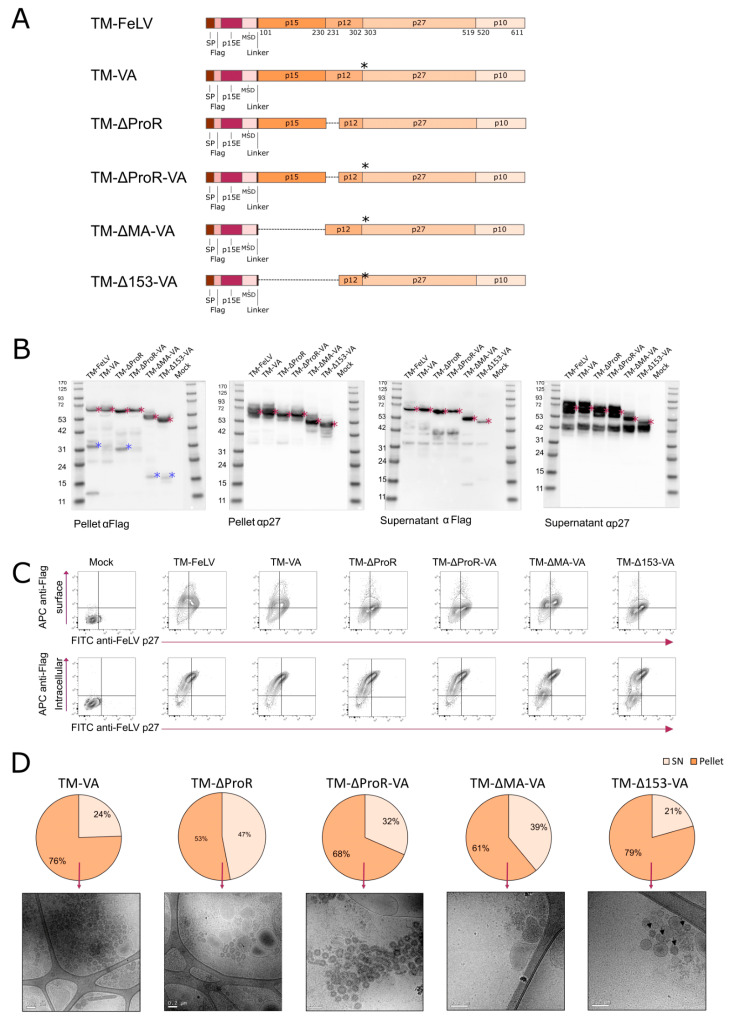
**Design and characterization of TM-Gag fusion proteins with altered Gag processing.** (**A**) Schematic representation of the modifications for each fusion protein. Black asterisks indicate L199V and R200A mutations. SP: Signal Peptide; MSD: Membrane-Spanning Domain. (**B**) Western blot developed with anti-Flag or anti-p27 antibodies to analyze the expression of TM-based fusion proteins in transiently transfected Expi293F cells. Red asterisks indicate the expected molecular weight. Blue asterisks indicate cleavage products. P: Cell Lysate; SN: Supernatant. (**C**) Representative flow cytometry panels for expression of the fusion proteins. Identification of intracellular Flag Tag and Gag. (**D**) p27 quantification of cell lysates and supernatants of the transiently transfected Expi293F cells. Cryo-EM images of the extracted and purified VLPs from the cell pellet are shown.

**Figure 5 ijms-24-09025-f005:**
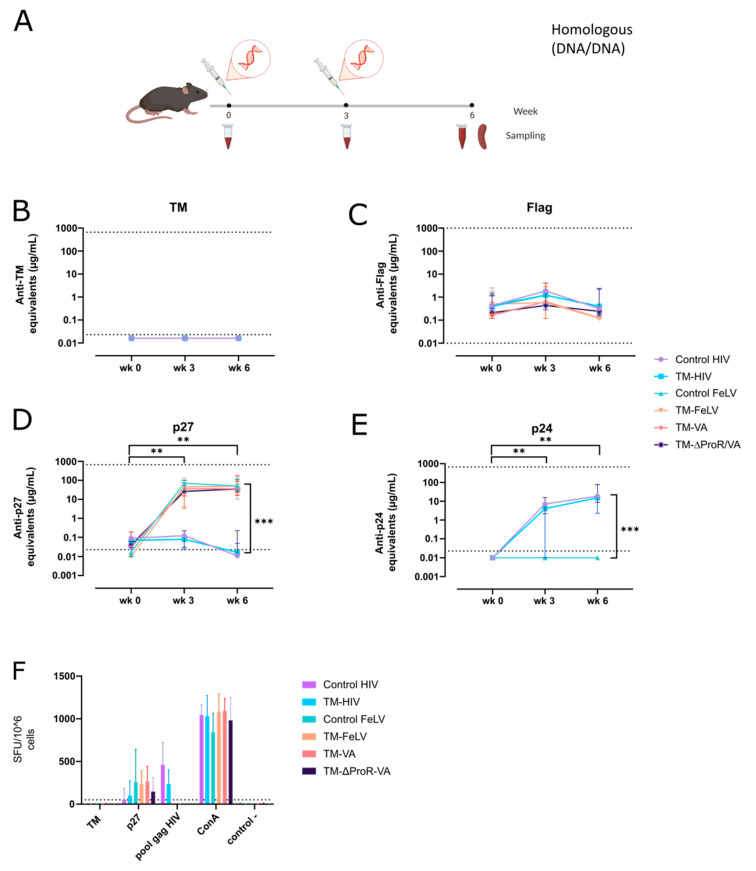
**Immunogenicity of TM-FeLV VLPs in C57Bl/6.** (**A**) Graphic scheme of the experimental procedure. Drawings were generated using Biorender.com. C57Bl/6 mice were electroporated twice with 20 μg of DNA. Antibody titers against TM (**B**), Flag (**C**), p27 (**D**), and p24 (**E**) are shown. Data is presented as the median with 95% CI of antibody concentration in mouse sera. Statistical differences were found using the Peto-Peto rank test for week 6 and the Paired Prentice-Wilcoxon test for longitudinal comparison (** *p* < 0.01; *** *p* < 0.001). (**F**) T-cell responses against TM, p27, and p24 were evaluated by IFNɣ ELISpot. Data is presented as mean with SD.

**Figure 6 ijms-24-09025-f006:**
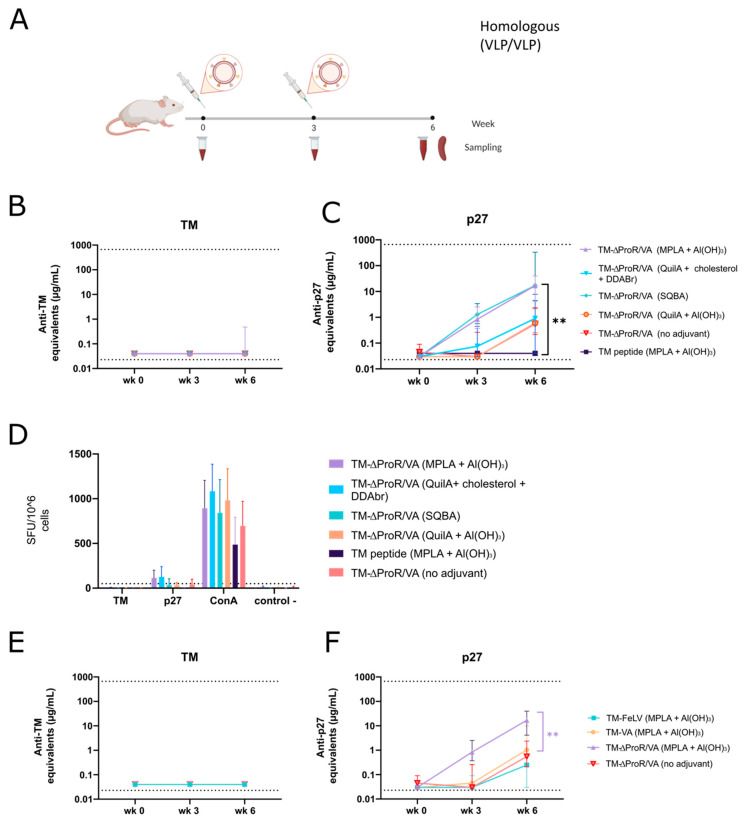
**Immunogenicity of TM-FeLV VLPs in BALB/c mice.** (**A**) Graphic scheme of the experimental procedure. BALB/c mice were immunized twice with purified and adjuvanted VLPs. Created with BioRender.com. Antibody titers against TM (**B**) and p27 (**C**) are shown. Data is presented as the median with 95% CI of antibody concentration in mouse sera. Statistical differences were found using the Peto-Peto rank test for week 6 (** *p* < 0.01). (**D**) T-cell responses against TM and p27 were evaluated by IFNɣ ELISpot. Data is presented as mean with SD. Humoral responses against TM (**E**) and p27 (**F**) when immunizing with different VLPs adjuvanted with MPLA + Al(OH)_3_ are shown. Data is presented as the median with 95% CI of antibody concentration in mouse sera. Statistical differences were found using the Peto-Peto rank test for week 6 (** *p* < 0.01).

## Data Availability

The data presented in this study are available on request from the corresponding author.

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
