# Peer review of "Exploring FeLV-Gag-Based VLPs as a New Vaccine Platform—Analysis of Production and Immunogenicity"

_ijms, 2023, doi:10.3390/ijms24109025_

Round 1

Reviewer 1 Report

This work describes a diligent effort to adapt an HIV-based VLP vaccine platform for the feline leukemia virus, and specifically to elicit an immune response against a transmembrane domain 15E, which is thought to provide the best chance at a neutralizing immune response. Ultimately it is not successful, but the failure of multiple strategies to produce extracellular recombinant VLPs is instructive and important to report. 

Questions about strategy that seem not to have been addressed (I might have missed it). (1) Why were the immunizations of the C57BL/6 mice performed by electroporating plasmid? Isn't the value of VLPs that they self-assemble in vitro? (2) Is it possible that testing the constructs in a feline cell line might have produced a different outcome?

Comments on clarity/ spelling:

1) The reference to the supplementary figure 1 and the labeling on that figure are not quite clear. Could you reference in the text which alternative SP is illustrated - GMCSF or immunoglobulin?

2) lines 238-240: the 30 kD and 20kD band patterns referred to in the text are not apparent to me. They seem to be in all lanes or no lanes, not in only two?

3) Lines 266, 306, and others: spelling error titters--> titers

Reviewer 2 Report

This innovative study by Ortiz et al describes a novel VLP platform using the gag protein of FeLV rather than HIV. They perform extensive work to optimize the VLP construct and overcome many technical hurdles in creating VLPs expressing the correct TM protein. They then attempt to utilize either the DNA construct or the VLP as an immunogen in C57/B6 and Balb/c mice. While they do see robust responses against FeLV gag (p27) they do not detect any responses by either ELISA or IFNg Elispot to the TM protein. 

This work is important as I did not know of any published studies which have utilized murine models to query the immunogenicity of the TM portion of FeLV. In literature review as part of this review process I also could not find any studies utilizing mice to evaluate the immunogenicity of the TM protein of FeLV. 

The negative findings in the murine studies then may just be that these two mice models do not have TCR repertoires that allows for recognition of the presented peptides of FeLV TM protein. This should be highlighted in the discussion section. This hypothesis could be confirmed by immunization of either rabbits (citations 20 and 42 in their work) or rats (DOI: 10.1159/000318892) to assess immunogenicity.

I do not believe this should be a barrier to publication, but rather a limitation that should be addressed with planned experiments for future publications. in animal models that are known to have responses to FeLV TM protein such as the papers mentioned above.  

All other suggestions are listed below:

-Line 306 titters is a typo and should be titers.

-For Figure 4C please include your extracellular anti-Flag staining flowplots for comparison to your other TM mutants.

-Similar to my above comments: what was the positive control for Anti-TM equivalents (ELISA in 6B and 6E) the negative data would be more convincing with a shown positive control to ensure the assay is working.

-Please include clonal ID for your antibodies in Methods (lines 505-507). Please also include your incubation times and temperatures to assist with reproducibility.

-What device was used for IM electroporation? I feel this would also be important for reproducibility.

-Where were the adjuvants purchased from and can they approximate mass equivalents as well as volumetric equivalents to help with reproducibility.

Reviewer 3 Report

Major comment:

Vaccines are important to induce humoral and cellular immunity that is effective in protecting against infection. In this study, the authors developed a Gag-based vaccine for FeLV. However, as the authors mentioned in this manuscript, the key antigens that induce neutralizing antibodies useful for infection protection are the two proteins gp70 and p15E that consist of the envelope.

In the present study, the authors designed various types of virus-like particles and analyzed their expression, confirming that the particles are not released outside the cell, but are accumulated inside the cell by TEM and western blotting. This would suggest that these particles likely do not retain the envelope protein, p15E in this study, an antigen necessary for neutralizing antibody induction. This is also suggested by the results of experiment on mice, which support that they cannot be used as a VLP vaccine.

In addition, while antibody induction against p27 and p24 was confirmed when used as a DNA vaccine, antibody induction against p15E was not confirmed, and antibody induction against Flag was very weak. Therefore, in this study, it is necessary to first collect basic information, such as properly evaluating whether p15E is expressed in 293 cells in the case of either plasmid, and evaluating whether VLPs contain p15E by Western blotting on purified VLPs.

Minor comments:

Why did you select p15E, not gp70?

In Fig 1D, positive and negative cells were clearly divided in staining with KC57. However, in staining with anti-p27 antibody, cells seemed not to be clearly divided. What is the difference between them?

It is better to change KC57 to anti-HIV p24 for clearly understanding.

How to determine the dose of DNA vaccine and VLP to mouse and why did not evaluate dose dependency?

Round 2

Reviewer 3 Report

Western blotting analysis of cells or supernatants is insufficient to evaluate whether p15 is included in the VLPs, because anti-Flag could be positive when p15 is expressed even if not present in VLPs. Therefore, it is necessary to confirm the presence of p15 in VLPs by immunoprecipitation, western blotting of VLPs purified by sucrose gradient method, or ELISA to detect p15 by capturing with anti-p27.

Round 3

Reviewer 3 Report

I understood the authors' reply.

There is no additonal comment.